# Locomotor Coordination, Visual Perception and Head Stability during Running

**DOI:** 10.3390/brainsci10030174

**Published:** 2020-03-18

**Authors:** Joseph Hamill, Jongil Lim, Richard van Emmerik

**Affiliations:** 1Biomechanics Laboratory, University of Massachusetts Amherst, Amherst, MA 01003, USA; 2Motor Control Laboratory, Texas A&M University – San Antonio, San Antonio, TX 78224, USA; jlim@tamusa.edu; 3Motor Control Laboratory, University of Massachusetts Amherst, Amherst, MA 01003, USA; rvanemmerik@kin.umass.edu

**Keywords:** perception, action, coordination, impact shock, shock attenuation

## Abstract

Perception and action are coupled such that information from the perceptual system is related to the dynamics of action in order to regulate behavior adaptively. Using running as a model of a cyclic behavior, this coupling involves a continuous, cyclic relationship between the runner’s perception of the environment and the necessary adjustments of the body that ultimately result in a stable pattern of behavior. The purpose of this paper is to illustrate how individuals relate visual perception to rhythmic locomotor coordination patterns in conditions during which foot–ground collisions and visual task demands are altered. We review the findings of studies conducted to illustrate how humans change their behavior to maintain head stability during running with and without various degrees of visual challenge from the environment. Finally, we show that the human body adapts specific segment/joint configuration and coordination patterns to maintain head stability, both in the lower extremity and upper body segments, together with an increase in coordinative variability. These results indicate that in human locomotion, under higher speed (running) and visual task demands, systematic adaptations occur in the rhythmic coupling between the perceptual and movement systems.

## 1. Introduction

Locomotor analyses have long been studied in biomechanics in both laboratory and non-laboratory environments [1,2]. The rhythmic patterns underlying locomotor behavior in mammals, including humans, are in part generated by the activity of neuronal Central Pattern Generator (CPG) circuits within the spinal cord [3,4]. These CPGs are thought to maintain the steady-state rhythms of walking and running patterns, as well as transitions between these patterns. However, another important cyclic or rhythmic relationship exists between the perceptual and locomotor systems, but less is known about this [5]. It has been suggested that individuals perceive their environment in terms of their ability to perform actions such as locomotion [6]. The relationship between perception and movement has been developed in what is known as ‘Perception–Action Coupling’ (see Figure 1). This coupling involves a continuous, cyclic relationship between information from the body (proprioception) and the environment and the necessary adjustments of the body from which a stable pattern of behavior occurs. Thus, perception and action (movement) are coupled to regulate behavior adaptively [7].

Prior studies on postural control have assessed changes in this Perception–Action Coupling under different visual task constraints [8,9,10]. These studies showed that postural control is not an autonomous system but part of an integrated action-perception system in which postural fluctuations and rhythms change as a function of the difficulty of the visual task imposed. Successful and safe navigation through our environment requires visual orientation to and active exploration of the surfaces of their environment [5,11] and sensitivity to the layout and nature of these surfaces (e.g., level, uphill or downhill, slippery). During steady-state locomotion, head ‘stabilization’ affords a consistent visual base during ambulation [12,13]. Head stabilization refers to the maintenance of head-in-space equilibrium [14] and reflects an important optimization for vestibular and visual function during locomotion [13,15].

There are several reflexive mechanisms contributing to the stabilization of the orientation of the head and eyes and these mechanisms have been well documented. The vestibulo-colic reflex (VCR) assists the stabilization of the head with respect to space by activating the neck muscles [16]. Head vertical translation during locomotion, for example, is compensated by counterbalancing head pitch rotations through the VCR. The stretch reflex of the neck muscles, the cervico-colic reflex (CCR), also stabilizes the position of the head with respect to the trunk [15]. The relationship between these two reflexes is complex; however, in general the VCR and CCR work together to stabilize the head in space [17]. Compensatory reflexive eye movements in response to head rotations, known as the vestibulo-ocular reflex (VOR), function to stabilize images on the retina and are also a part of a gaze stabilization system. However, these reflexive adjustments are limited and only work appropriately when head rotational velocities are under 350°/s [18]. Furthermore, the rotational VOR gain defined as the change in the eye angle divided by the change in the head angle during the head turn (ideal at 1.0) can be as low as 0.75 during running [19]. These findings suggest that in order to account for the deficiencies in visual stabilization as well as to reduce the demands made of the reflexive adjustments, additional active and passive regulations must be employed for the stabilization of the head in space.

During locomotion, the foot–ground contact, particularly in running, results in a shock wave that is transmitted throughout the musculoskeletal system. This shock wave is propagated towards the head, reaching the head approximately 25 ms after the foot–ground contact with the result that it may disrupt head stability [20]. Accordingly, the musculoskeletal system must adapt to attenuate the shockwave as it passes along the body. The foot ground collision and resulting shock wave occurs in the 10–20 Hz range in a frequency analysis of the impact [21]. The individual must attenuate the higher frequencies of the shock wave such that the shock is in the region less than 10 Hz before it reaches the head. The result of the adaptation of the total system is that the accelerations that reach the head subsequent to the foot–ground collision in locomotion show little change across different locomotor conditions [20]. Since the acceleration of the head does not vary significantly in many different locomotor conditions remaining at approximately 1 g, the result is that the head may be considered stable during locomotion. This appears to be an important criterion during locomotion. Researchers have suggested that minimizing head movement (i.e., head stability) is critical in maintaining a relatively constant visual field allowing the individual to continue a visual reference [13].

One possible way in which the locomoting individual may adopt to maintain head stability is to alter their coordination patterns. Coordination of different segments of the body has been investigated in locomotion studies to understand the organization of segments from a dynamical systems perspective [22,23]. The coordination of segments or joints has been studied by coupling contiguous oscillators (i.e., segments or joints) using one of two methods: 1) continuous relative phase (CRP), indicating both a spatial and temporal relationship of the coupling [24]; or 2) modified vector coding (mVC), indicating a spatial orientation of the coupling [23]. The latter method used the most often. If necessary, a runner can adopt a new coordination pattern to maintain head stability if head stability is perturbed. Examining the coordination patterns and the variability of these patterns during locomotion can provide information about the organization and flexibility of locomotor patterns in response to the perceptual task demands.

The purpose of this paper is to use the Perception–Action Coupling perspective to illustrate how individuals relate visual perception to rhythmic locomotor coordination patterns in conditions during which foot–ground collisions may be altered. We will consider three studies that were conducted in our laboratory that attempted to answer the questions relating visual precision during altered stride frequency conditions (i.e., changing stride length to cause different foot–ground collisions). The first study addressed the issue of providing individuals feedback of their head/gaze orientation and the resulting effect on shock attenuation. The second study identified adaptive changes in running kinematics and impact shock transmission as a function of head stability requirements while the third study identified change in coordination with increasing head stability requirements.

## 2. Shock Attenuation and Head Stability

Head stability has been evaluated using a number of different methods [13] including using typical phase-plane trajectories of the head and joint displacements [25]. Holt and colleagues used the consistency of the endpoint trajectory and the segmental motions. They suggested that, from a dynamical systems approach, preferred behaviors emerge from stability considerations. In this scenario, as the person learns to walk for example, there is an eventual discovery of the most stable state and a tendency to remain in that state. All of the participants were reported to have a minimum standard deviation for the vertical head trajectory at the preferred or predicted frequency. That is, the individual may explore the dynamical space until honing in on the most stable state for the head. Overall, these researchers concluded that head stability is crucial to skilled performance. Internally generated perturbations on the visual-vestibular system may make dealing with externally generated perturbations more difficult. Information from these systems could, potentially, become unreliable and lead to control problems. However, in this study, the possible perturbation resulting from the foot–ground collision was not considered.

As mentioned previously, with every foot–ground collision, a shock wave passes through the musculoskeletal system from the impact point to the head [20]. The transmission of this shock to the head, however, is attenuated regardless of the leg impact force during normal locomotion. Potentially, this shock wave may disrupt the stability of the head and thus the visual field. Hamill et al. [20] assessed the relationship between stride frequency and shock attenuation in running. They constrained healthy participants to increased stride frequencies above (+10% and +20%) and below (−10% and −20%) of their preferred stride frequency (PSF) at preferred running speed. At the higher stride frequencies, the resulting impact measured at the tibia was observed to be much less than at the lower stride frequencies. They also reported that, regardless of the impact at the tibia (approximately ranging from 6 to 8 g), the impact was attenuated such that the head impact was relatively constant at approximately 1 g [20,26] (see Figure 2).

How the system attenuates this impact shock to maintain head stability has been of great interest in the literature. It has been suggested that the main locus of the attenuation takes place during knee flexion in the initial portion of the stance phase [21,27]. Derrick et al. [21] used a Newton–Euler inverse dynamics procedure to calculate joint moments and power and accelerometry data during running at different stride lengths to evaluate shock attenuation. The impact load on the body increased with increased stride length (reduced stride frequency) even at a constant running velocity. They concluded that energy absorption was greater at the knee joint than at the hip or ankle over all stride length conditions. When the foot–ground collision resulted in a greater impact shock, the runner increased their knee flexion angle.

Edwards et al. [27] came to a similar conclusion regarding shock attenuation, as did Derrick and associates [21], although using a much different method (inline skating which has no heel strike versus running which has a heel strike respectively). In this study, the researchers placed the participants in various degrees of knee flexion and imparted an impact shock to them. Shock attenuation between the leg and head was evaluated using accelerometry. They reported that a non-linear increase in shock attenuation was observed as the knee joint became more flexed. They concluded that the knee joint acts as a low-pass filter with a variable cut-off frequency, allowing greater shock attenuation with increased knee flexion. They suggested that increasing knee flexion angle may shift the shock attenuation responsibilities away from passive tissues and toward active muscle tissue. Attenuation by active muscle tissue allows fewer of the high frequencies present in the impact to be transmitted to the head and thus improves shock attenuation and maintain head stability.

In support of this view, two recent studies [28,29] demonstrated an association between the changes in visual orientation and shock attenuation at the head during running, which presumably resulted from different levels of neck muscle activation. When visual orientation was constrained by asking runners to look at the different visual target locations in pitch (or the sagittal plane) [28] and yaw (or transverse plane) [29], greater shock attenuation between tibia and head was observed when the head was more flexed (e.g., looking down) or was at the rightmost directions (e.g., 90° rotation to the side). Although the neck extensor muscle activation levels were not directly measured in these studies, it has been known that muscular activity in the neck muscles increases with greater neck extension and rotation [30] and when the head moves out of its neural posture [31]. Further, the previous literature has consistently shown that neck muscle activities are able to compensate for changes in the motion of the trunk to maintain head position and the eye-fixation point relative to the trunk [32,33]. Further studies seem warranted to examine the relationship between the activation of the muscular system at different body sites, in particular the head, neck and trunk, and shock attenuation during running.

## 3. Head Stability, Shock Attenuation and Visual Feedback

Human locomotion is often described as self-optimizing, thus producing stable coordinated patterns that are also energy efficient [34,35]. The optimization of these patterns affords the selection of safe and efficient paths of navigation [5,36]. Under steady-state conditions without secondary tasks, individuals select movement patterns that minimize the amount of shock transferred to the head. Specifically, modulation of stiffness [27], kinetics [21] and muscle activity [37] appear to be the mechanisms through which individuals modulate the amount of high-frequency shock transmitted to the head, and thereby possibly stabilize the visual field by reducing head motion [20,21]. If we look at the impact acceleration in the frequency domain (see Figure 3), frequencies above 10 Hz represent the impact shock of the foot–ground collision. This area is often referred to as the ‘passive’ region. Below 10 Hz is the portion called the ‘active’ region. That is, this region is under active control by the musculoskeletal system.

Investigations into walking have shown that individuals display stable head acceleration patterns across a wide range of step lengths, cadences, and speeds as well as a combination of stride lengths and cadences [38]. The stabilization of head accelerations across a range of speed and stride characteristics results from modifications of the movement kinetics, kinematics and muscular activation patterns and may afford a stable visual field and the identification of salient visual information and safe navigation through their environment [39].

The modulation of gait patterns to minimize shock transmission through the kinematic chain and to regulate head acceleration is a critical factor in the successful navigation through complex environments [5,40]. Previous studies have shown that real-time visual [41] or auditory [42] feedback can reduce tibial accelerations. It is not clear, however, whether the minimization of impact shock in response to the aforementioned feedback stimuli results in the anticipated changes in head accelerations. Providing real-time feedback of head/gaze motion while running at a range of stride frequencies at preferred speed may allow a direct assessment of how tibial and head accelerations and shock attenuation are modulated in responses to different visual tasks.

Busa et al. [43] sought to investigate tibial and head accelerations during running when individuals were provided with visual feedback of their head/gaze orientation. They employed a rather unique experimental setup (see Figure 4). The experimental protocol (i.e., running conditions) was similar to that of Hamill et al. [20]. The results of this study supported previous findings that lower stride frequencies result in greater shock attenuation and a generalized stabilization of head accelerations [21,37,44]. Head and tibial accelerations resulting from initial foot–ground impact appeared to be associated with stride frequency but not with the nature of the visual task imposed. The reduction of head acceleration during the active portion of stance (e.g., increased muscle activation) when individuals were provided feedback of their head/gaze orientation appeared to be the mechanism by which head/gaze motion is reduced. These results suggested that, although the initial foot–ground impact was not changed, the control of head motion and the regulation of head accelerations during preferred speed was task dependent, such that individuals adopted movement patterns that reduce head accelerations and stabilize head/gaze motion during the (late) active phase of stance.

## 4. Adaptive Changes as a Function of Head Stability

Previous studies on running have demonstrated modulation of shock transmitted through the body across a range of impulse loads, resulting in stable head accelerations [26]. Increased shock attenuation has been observed in response to changes in gait parameters [20,21,45]. The stabilization of the head in response to altered stride parameters and running speed has been well established. The inverse, however, how the body adapts and attenuates impact shock to different visual tasks or head stability demands is less clear. A study by Lim et al. [46] addressed this issue. Using an experimental setup similar to that of Busa et al. [43], these researchers investigated how individuals adapt their running kinematics and alter their impact shock transmission in response to increasing head stability requirements. Here, the participants ran at the preferred running speed under seven different visual conditions while they received real-time feedback of the intersection point of their head/gaze on the frontal plane (see Figure 4b). Additionally, a static box was projected onto the same screen. Participants were instructed to look ahead and keep the head/gaze point within the box. Seven different box sizes were used, ranging from 21° to 3° of visual angle (VA) in 3° decrements (VA-21, VA-18, VA-15, VA-12, VA-9, VA-6 and VA-3, respectively). The seven visual conditions resulted in increased task difficulty from VA-21 to VA-3 (see Figure 5).

With increased head stability demands, these researchers reported that, during the active phase of stance, peak acceleration and signal power of the head decreased. During the impact portion of stance, tibial and head acceleration as well as the transmission of shock did not change with different visual conditions. The observed changes in head dynamics during late stance coincided with altered lower limb kinematics at impact and toe-off as well as reduced range of hip and knee motion (see Figure 6). Overall, the most difficult visual task (e.g., VA-3) resulted in a 32% reduction in head/gaze motion and an 8% increase in stride frequency. These mechanisms employed in stabilizing head/gaze during running likely play an important role in how individuals successfully optimize the system to navigate complex terrain and select safe and efficient paths [47]. 

This particular study provided strong empirical evidence that individuals can tune their running mechanics in order to stabilize head orientation under increasing visual task demands. The mechanisms by which individuals stabilized the head with increased visual task difficulty appear to predominantly occur in late stance suggesting these are active adjustments. The observed changes in stride parameters indicate individuals increase stride frequency (and decrease stride length) while preserving stance time and decreasing aerial time. Additionally, changes in lower limb joint configurations appeared to be adaptive to maintain shock transmission to the head and reduce the vertical center of mass displacement, both of which serve to increase head stability in line with task demands. From this study, individuals make subtle, but important, changes in running mechanics and stride parameters that optimize head/gaze stability in response to increased visual task constraints.

## 5. Coordination and Head Stability

An apparent goal of the human system during locomotion is to maintain a stable visual field in light of the perturbations that occur during the cyclical actions of locomotion and physical interaction with the environment. However, due to the high demands on multiple sensorimotor systems as well as multi-segmental coordination requirements across the whole body, understanding the exact mechanisms that contribute to attaining head stability in locomotion is not a simple process. As such, based on the viewpoint that the whole body is an integrated head/gaze stabilization system [12], previous literature has examined how several subsystems, including eye, head, neck, trunk, and lower limbs, are coordinated with each other to stabilize the head/gaze system and, therefore, maximize the quality of visual information acquired from the environment.

Indeed, previous studies on walking have demonstrated several mechanisms of the subsystems in facilitating head stability. During normal walking, healthy individuals maintain a high degree of head stability through compensatory movements such as adjustments in head pitch that counteract linear and angular motions imposed by the whole body [13,48]. More specifically, Cromwell and colleagues found that compensatory (or equal and opposite) head-on-trunk movement with respect to trunk ensured head stability in both the horizontal [14] and sagittal [49] planes during walking. These mechanisms appear to work for a wide range of the frequency spectrum of trunk movements. However, a tighter control of head-trunk coordination (e.g., improved compensation of head-on-trunk movements) via feedforward mechanisms was suggested for head stability in challenging conditions (e.g., high frequency perturbation at the trunk motion).

Additionally, a number of studies in walking and running highlighted the head stabilization mechanisms in response to altered gait speed and stride parameters. Hirasaki et al. [50] demonstrated that the motor system appeared to be optimized to maintain head/gaze over a wide range of walking speeds although the compensatory mechanisms for the stability of head orientation were different over the range of walking speeds. During slow walking (below 1.2 m/s), head pitch relative to the trunk was a dominant compensatory movement for trunk pitch, thus maintaining a relatively stable head position in space. However, as walking speed increased (between 1.2 and 1.8 m/s), head pitch rotation in space tended to compensate for trunk vertical translation while exhibiting increased head translation with invariant trunk pitch. Walking speed-related contribution for the enhancement of head stabilization was also exhibited between the trunk and neck segment. Kavanagh et al. [51] revealed that while the trunk segment plays a role in regulating oscillations in all directions, additional control from the neck segment was required as walking speed increased, but only in the direction of travel (i.e., anterior–posterior direction). In line with these observations, the best overall stability of head and pelvis accelerations was achieved when preferred step length and cadence and, therefore, walking speed were employed [38]. Similarly, in running, an association between lower extremity and pelvis kinematics and the upper trunk and head stability has also been demonstrated [52,53].

These prior studies provided evidence of the existence of multi-level compensatory mechanisms residing in the system and demonstrate key biomechanical contributors and their significance in stabilizing the head during locomotion. However, joint/segment coordination patterns that underlie the compensatory adjustments are not well understood, especially during running. Coordination and coordination variability have been used to assess both the spatial and temporal organization of movements as well as runner’s adaptability in response to changes in constraints [54,55]. Among the various coordination analysis techniques, vector coding (VC) has commonly been employed in the locomotion literature. Modified VC allows easier interpretation of biomechanical running data [56] and provides useful metrics in understanding the organization of movements such as phase dominancy (e.g., in-phase or anti-phase) and segmental dominancy based on spatial changes [57,58]. However, currently, modified VC has not been used to assess how coordinative changes contribute to stabilize the head during running.

To understand how the greater head-in-space equilibrium during running is achieved through movement coordination, coordination patterns and variability analyses of Lim et al. [46] were performed between head and trunk segments, hip and knee joints, and knee and ankle joints in the three cardinal planes using modified VC. The observed changes in coordination patterns indicated that a joint/segment-specific function may exist for the stabilization of head motion during running. This stabilization was accomplished by reducing the vertical range of motion through the lower extremity coordination (e.g., increased in-phase and decreased anti-phase sagittal plane coordination) (Figure 7a,b) and fine-tuning of the rotational control in the trunk and head (e.g., increased anti-phase transverse plane head–trunk and head-leading coordination patterns) (Figure 7c). The mechanisms that individuals regulate in both upper and lower body appear to occur in late stance, emphasizing active (low frequency) coordinative adjustments. Additionally, an increase in coordination variability in both the upper body and lower extremity couplings may serve to enhance the flexibility of the system in line with increasing visual task demands (Figure 8). Overall, the results demonstrated that runners tune their coordination patterns in order to meet increasing head stability demands.

## 6. Summary

Based on the concept of Perception–Action Coupling, perception is both generated by movement and provides information for action. In the words of James Gibson [6], from whom the notion of Perception–Action Coupling stems, "We must perceive in order to move, but we must also move in order to perceive". Thus, an important cyclic or rhythmic relationship exists between the perceptual and locomotor systems. When an individual moves, the perceptual information gained is used to regulate the dynamics of the moving system to control or adapt the resulting movement. Prior studies have assessed changes in the Perception–Action Coupling under different visual constraints indicating that postural control is an essential part of an integrated action-perception system. During steady-state locomotion, head stabilization is a priority in maintaining a consistent visual field. While there are several reflexive mechanisms that contribute to head stabilization (e.g., the vestibulo-colic reflex, the cervico-colic reflex, and the vestibulo-ocular reflex), other active and passive actions must be employed to stabilize the head in space. Changes in coordination mechanisms can also attenuate the impact from foot–ground collisions. During the stance phase of locomotion, a shock wave caused by a foot–ground collision can also be attenuated by increased knee flexion of the support limb or increasing stride frequency.

In this paper, we reviewed three studies from our laboratory investigating the role of vision during locomotion during the ground contact phase. The foot–ground collision results in a shock wave that is ultimately transmitted to the head that possibly causes a disruption in head stability. The first study gave visual feedback to the participants while altering stride frequency and reported that, consistent with previous studies, head acceleration was maintained, mainly as a result of active control by the musculoskeletal system, at a steady state regardless of the magnitude of the impact [43]. In the second study, under conditions in which there was a visual challenge and altered foot–ground impact, it was found that the most difficult visual task resulted in a reduction in head/gaze motion and an increase in stride frequency [46]. In the last study, it was considered that one possible way that the locomoting individual could adopt to maintain head stability during visual challenges was to alter their coordination patterns [59]. It was shown that changes in coordination patterns indicated that a specific coordination alteration was accomplished to maintain head stability. In the lower extremity, there was a decrease in the anti-phase movements in the sagittal plane and decreased trunk involvement in the transverse plane. There was also increased head contribution to the coordination changes. That is, the individual can ‘tune’ their coordination patterns to meet increasing head stability demands.

The studies presented here demonstrate that action and perception are intimately related and that visual feedback of head movement results in systematic adaptations or ‘tuning’ in coordination and control during the late or active phase of stance during running. This modulation of gait patterns to minimize shock transmission through the kinematic chain to regulate head acceleration is a critical factor in the successful navigation under more stable and predictable conditions in which human locomote; however, they may also underlie the adaptations needed in more complex environments that require greater modulation of the Perception–Action Coupling. 

In our laboratory studies, all participants ran on a treadmill, which is a reasonable surrogate for over-ground running. However, the perception of speed, movement or optic flow may not be the same. Therefore, repeating our laboratory studies over ground rather than on a treadmill may present a more ecological setting for interpreting the coupling of movement and perception and the control of head stability. In addition, the methods and techniques presented in this review can be applied to those with visual impairments and/or gait asymmetries, where different demands on head control may exist or a diminished capacity to modulate coordination patterns in response to different visual task constraints.

## Figures and Tables

**Figure 1 brainsci-10-00174-f001:**
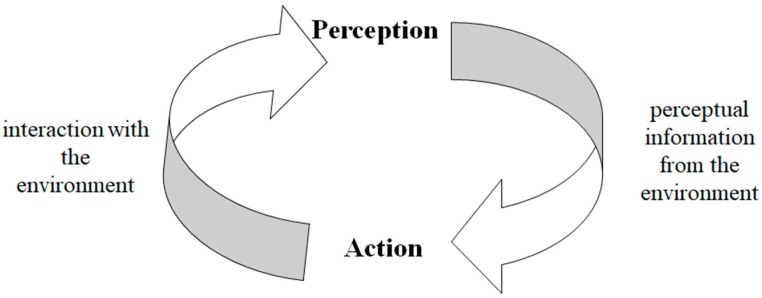
A schematic illustrating Perception–Action Coupling.

**Figure 2 brainsci-10-00174-f002:**
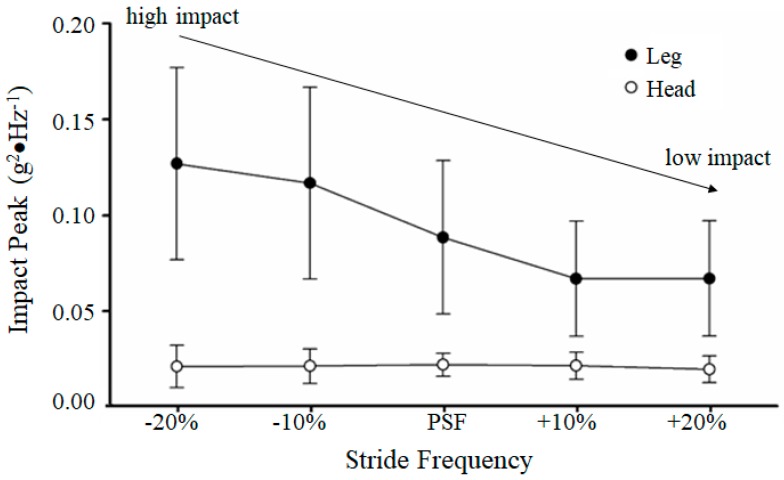
Mean (±SD) of impact peak of tibia and resulting peak of the head during the foot–ground collisions at the preferred stride frequency (PSF) and plus and minus percentages of the PSF (adapted from data in Hamill et al. [20]).

**Figure 3 brainsci-10-00174-f003:**
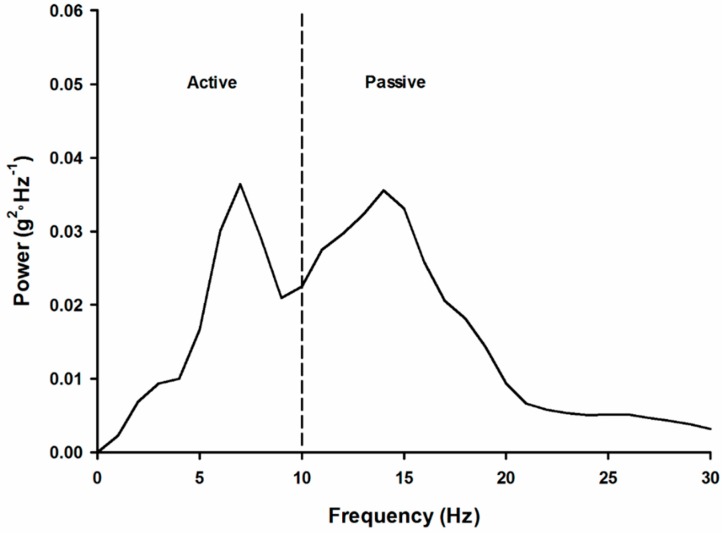
Tibial acceleration profile in the frequency domain.

**Figure 4 brainsci-10-00174-f004:**
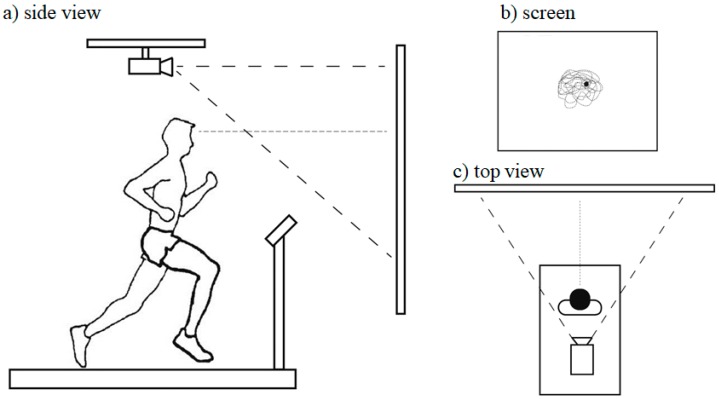
Experimental setup as in Busa et al. [43]. Kinematic data were collected along with tibial and head accelerations. The participants ran at their preferred speed and percentages of their preferred stride frequency. A screen was positioned approximately 2.5 m away from the treadmill center. The dotted line from the participant’s head in (**a**) and (**c**) indicates an imaginary line of a head gaze vector created from the motion capture data of the head. In the visual feedback condition (**b**-right), a dot indicating the intersection point of head gaze vector on the screen was displayed while running. In (**b**) the dotted line represents the trajectory of the head gaze point on the screen, this was not displayed during the testing. During the visual feedback condition (**b**), a square box (inset light box) was displayed to indicate the boundary area for the feedback. This box was created by subtending an angle 21° horizontally and vertically form the center of the treadmill centered at a height of 1.7 m above the treadmill belt.

**Figure 5 brainsci-10-00174-f005:**
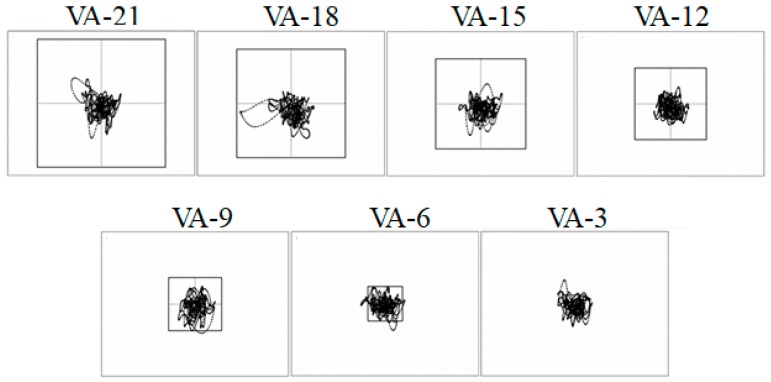
A schematic of the square boxes representing the visual angle areas in which the head projection vector had to be maintained. The angles subtended by the box horizontally and vertically were varied in each condition from 21° to 3° of visual angle with 3 degree decrements. An example of one participant’s head movement is illustrated (note that the participant did not see the trace of the path; they only saw the movement of the dot from the projection vector). Adapted from Lim et al. [46].

**Figure 6 brainsci-10-00174-f006:**
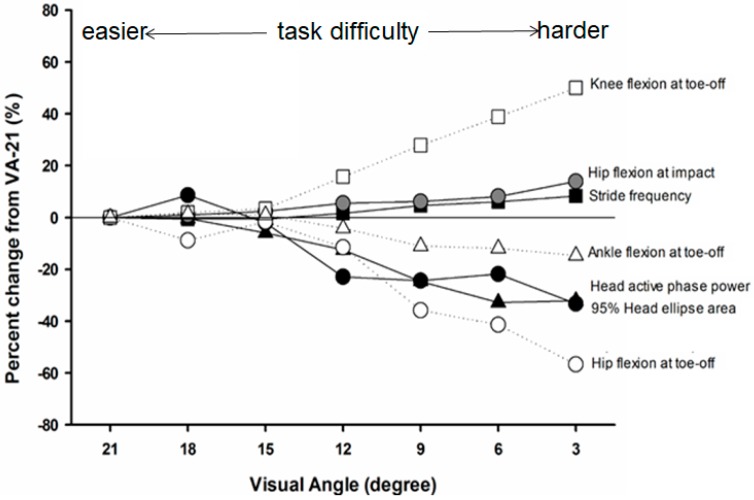
Mean percent changes for key dependent measures showing the significant effect of visual condition from VA-21. Positive numbers indicate increased value in percentage from VA-21, and vice versa (adapted from Lim et al. [46]).

**Figure 7 brainsci-10-00174-f007:**
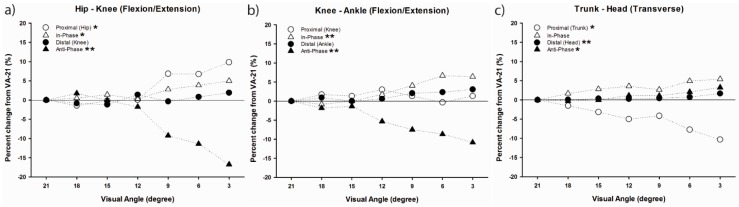
Mean percent change for the coordination patterns in lower extremity joint couples in the sagittal plane, hip–knee (**a**) and knee–ankle (**b**), and upper extremity segment couples in transverse plane, trunk-head (**c**), showing the significant effect of visual condition from VA-21. Main effect of visual condition (* *p* < 0.05; ** *p* < 0.01). Positive numbers indicate increased value in percentage from VA-21, and vice versa (Lim et al. [59]).

**Figure 8 brainsci-10-00174-f008:**
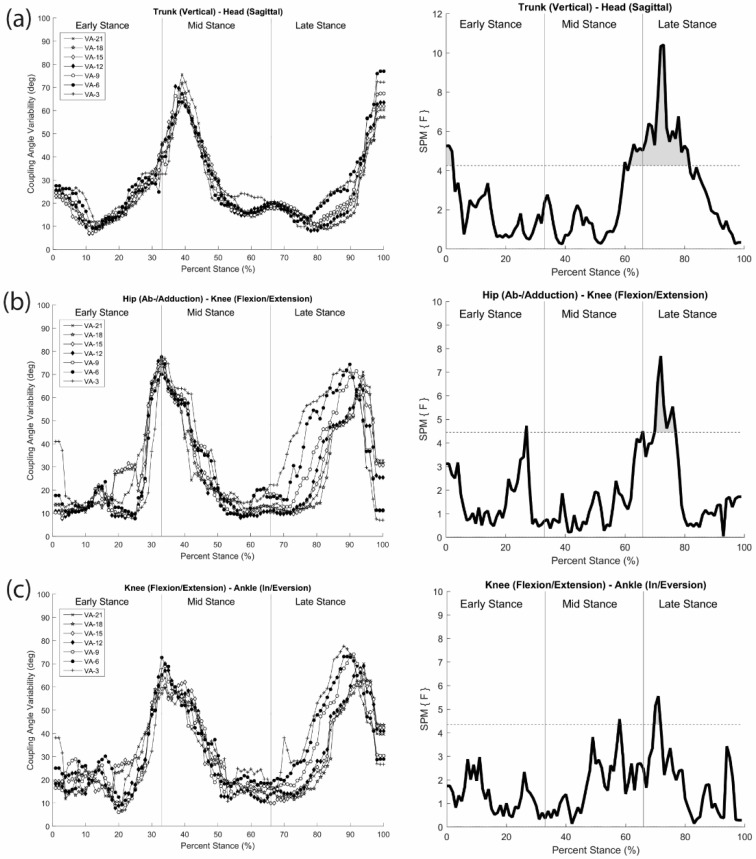
Coordination variability during stance in the head–trunk (**a**) and lower extremity (**b** and **c**) segment/joint couplings as a function of visual condition (left) and F (SPM{F}) values at each normalized time point (right). A horizontal dotted line in the right panel indicates threshold level in SPM statistics (Lim et al. [59]). Note. SPM indicates a statistical parameter mapping, providing a statistical solution for the objective classical hypothesis testing for differences between biomechanical time-series datasets (Pataky et al. [60]).

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
