# Peer review of "Locomotor Coordination, Visual Perception and Head Stability during Running"

_brainsci, 2020, doi:10.3390/brainsci10030174_

Round 1

Reviewer 1 Report

General comments:

The authors present a review on a current and pertinent subject; the role of vision in locomotion during the ground contact phase. The principles of Gibson’s perception-action were the theoretical support to explain how visual information plays a significant role in motion perception, and in head stability in particular. The authors considered three studies taking into account the relation of stride length and foot-ground collisions on head stability. The article is well written, pleasant to read, and the figures are phenomenal. One part that I would like to see more explored was when visual inputs are in conflict with other sensory channels about self-motion because it is not always the prevailing source of information.  

Specific comments:

Lines: 86-90: The authors stated that mVC is the most used method. It could be relevant to say way. Maybe because it incorporates both shape and magnitude, and it allows to measure joint angle velocity. Additionally, in the past, there were some critics regarding CRP.  Although the authors mentioned in lines 320 to 324, I would like the authors to include methodological advantages and disadvantages in this section of the article.

Lines 141-142: The authors compared ref [21] with [27] stating “… although using a much different method.”. The task is totally different. While ref [21] is about running, ref [27] is on inline skating. I think that it is relevant to report the differences in the task referring to common aspects (one requires heel strike, the other is about sliding).

Lines 155-159: I think that this paragraph could be more explored. The authors stated “…greater shock attenuation between tibia and head was observed when the head was more flexed…”. Then, an example with the sternocleidomastoid was given. I think that it is more relevant to report information about the back muscles of the neck. Despite the rotation of the head, If the head is flexed, the back muscles of the neck are performing an eccentric action.  

Line 280: What is the meaning of “… compensatory…”. Should be because there is LESS head-on-trunk movement comparing to the trunk movement? Explain better to the reader, please.

Line 337: The authors wrote “…upper (a) and lower extremity (b and c)”. It seems that they are referring to upper extremity when they are mentioning the Trunk-Head relationship.

Line 340: The authors should explore with a sentence the reason why Statistical Parameter Mapping is a useful tool for this kind of data, i.e., type of biomechanical data (with Bayesian inference or frequentist perspective).

Finally, I would expect some recommendations for future directions. For example, the perception of speed on a treadmill is faster when compared to overground; while the stride time over time can be persistent in both treadmill and overground, speed is anti-persistent when walking on a treadmill (relating non-linear measures). Or what is important know what should be explored about the relationship among reduced stance time, knee flexion, foot landing position, and stride frequency during the ground contact period.

Thank you!

Author Response

Reviewer 1

General comments:

The authors present a review on a current and pertinent subject; the role of vision in locomotion during the ground contact phase. The principles of Gibson’s perception-action were the theoretical support to explain how visual information plays a significant role in motion perception, and in head stability in particular. The authors considered three studies taking into account the relation of stride length and foot-ground collisions on head stability. The article is well written, pleasant to read, and the figures are phenomenal. One part that I would like to see more explored was when visual inputs are in conflict with other sensory channels about self-motion because it is not always the prevailing source of information.  

Specific comments:

Lines: 86-90: The authors stated that mVC is the most used method. It could be relevant to say way. Maybe because it incorporates both shape and magnitude, and it allows to measure joint angle velocity. Additionally, in the past, there were some critics regarding CRP.  Although the authors mentioned in lines 320 to 324, I would like the authors to include methodological advantages and disadvantages in this section of the article.

Modified Vector Coding (mVC) is the most common method used because it can be related back to the original data. This is not true for the other methods. We have decided not to include advantages and disadvantages since this would take away from the original intent of the paper. However, methodological advantages and disadvantages of coordination and coordination variability analysis were well addressed in [56].

Lines 141-142: The authors compared ref [21] with [27] stating “… although using a much different method.”. The task is totally different. While ref [21] is about running, ref [27] is on inline skating. I think that it is relevant to report the differences in the task referring to common aspects (one requires heel strike, the other is about sliding).

We have added that one study was on inline skating while the other was on running (Lines 143-144).

Lines 155-159: I think that this paragraph could be more explored. The authors stated “…greater shock attenuation between tibia and head was observed when the head was more flexed…”. Then, an example with the sternocleidomastoid was given. I think that it is more relevant to report information about the back muscles of the neck. Despite the rotation of the head, If the head is flexed, the back muscles of the neck are performing an eccentric action.  

We agree with reviewer’s comments. The example of ‘sternocleidomastoid’ has been changed to ‘neck extensor muscle activation levels’ (Line 159).

Line 280: What is the meaning of “… compensatory…”. Should be because there is LESS head-on-trunk movement comparing to the trunk movement? Explain better to the reader, please.

The meaning of “compensatory head-on-trunk movement” was clarified by adding the terms, ‘or equal and opposite’ (Line 282).

Line 337: The authors wrote “…upper (a) and lower extremity (b and c)”. It seems that they are referring to upper extremity when they are mentioning the Trunk-Head relationship.

We have changed this to read ‘head-trunk (a) and lower extremity (b and c)’ (Line 339).

Line 340: The authors should explore with a sentence the reason why Statistical Parameter Mapping is a useful tool for this kind of data, i.e., type of biomechanical data (with Bayesian inference or frequentist perspective).

While we feel that the reviewer makes an important point regarding SPM, the thrust of the paper is not SPM and so the authors feel that by adding anything else regarding SPM would take away from the paper as a whole. However, a phrase addressing the significance of SPM in biomechanical datasets has been added (Lines 342-344).

Finally, I would expect some recommendations for future directions. For example, the perception of speed on a treadmill is faster when compared to overground; while the stride time over time can be persistent in both treadmill and overground, speed is anti-persistent when walking on a treadmill (relating non-linear measures). Or what is important know what should be explored about the relationship among reduced stance time, knee flexion, foot landing position, and stride frequency during the ground contact period.

We have added the following text in the summary section (Line 384-392): “In our laboratory studies, all participants ran on a treadmill which is a reasonable surrogate for over-ground running.  However, the perception of speed, movement or optic flow may not be the same.  Therefore, repeating our laboratory studies over-ground rather than on a treadmill may present a more ecological setting for interpreting the coupling of movement and perception and the control of head stability.  In addition, the methods and techniques presented in this review can be applied to those with visual impairments and/or gait asymmetries, where different demands on head control may exist or a diminished capacity to modulate coordination patterns in response to different visual task constraints.”

Reviewer 2 Report

Summary

This literature review summarizes results from studies investigating behavioral mechanisms used to optimise head stability during running in humans. The purpose of the study is to focus on how visual perception influences locomotor coordination and reduction of ground-related impacts as well as how changes in joint-segment coordination may influence head stability and visual perception. After an Introduction of the Perception-Action framework and demonstrating that reflexive mechanisms engaged in head stabilization may not be enough to explain the whole head stabilization process, authors describe results from three main studies. The main results from each study represents a whole section and is presented in perspective with results from other studies. A paragraph on how coordination contributes to head stability follows, and a final paragraph summarizes the knowledge we gain from this review.

General comment

The work done by the authors is very good. The paper is very well written, problematics and results are presented with insight and perspective. This review will surely be an important contribution to a developing field of research and may become an important reference paper in the future. I do, however, have two main comments and a few questions and recommendations.

Main comments

Authors did not explain how they selected the different papers they chose to present in this literature review. Did you select them on the basis of keywords? Which database(s) did you use? Were some papers excluded? If yes, can you provide the methodology that has been used to select or disregard papers from this manuscript?

It may be concerning that authors refer to data from a paper that is still “under review” (and therefore comes from the same research group). Although I trust the honesty of the authors for presenting these results, changes may be recommended by the experts currently reviewing the other paper. Wishing you the best of luck for publication of the other paper, there is no way for me to verify the results and design of the experiment of Lim et al. [59].

Remarks / recommendations

Since the authors chose the Action-Perception framework to position their work, I was wondering if the current title may not be misleading as authors do not refer to the perception concept. Biomechanicians may see something only referring to segmental coordination. Maybe consider including perception somehow?

In the Introduction, authors relate to perception as “individual’s perception of the environment” (l. 37-38) and to “perception-action coupling under different visual task constraints” (l. 42-43). Although it is clear that the authors’ point is to drive the reader towards visual perception that indeed contributes to perception of the environment, the whole perception mechanism is not limited to the environment nor to vision only. Perception of our own bodies may also influence our actions (and vice-versa), while several sensory mechanisms could be at work additionally to visual feedback. Several references to other sensory systems are actually made throughout the paper. Therefore, I only recommend adding a sentence or two to clarify this and then dive into what authors will focus on in the review. According to this remark, the schematic illustration of the Perception-Action coupling could also be clarified with something like “perceptual information from the environment and the human body” coming with the right-side arrow.

I recommend clarifying why authors chose to focus on the three specific studies they are presenting within individual sections and why they chose to separate each study from one another. At first glance, these three studies may look quite similar. One sentence or two to clarify the authors’ idea about why each study is different from another may help to understand the organization of sections 2, 3 and 4.

In section 2, l. 151-152, it is mentioned that studies from Lucas-Cuevas et al. [28] and Mangubat et al. [29] support the view and the results of Hamill et al. study [20]. Given the content of both these studies, it questions why authors decided the study from Hamill et al. was the “main” study in this paragraph. A little clarification may help the reader understand why the study from Hamill et al. is the main one referred in this paragraph.

l. 242, are authors talking about head stability demands or gaze stability demands? Given the different conditions presented in Lim et al. study, they constrained gaze orientation over head orientation. If both are obviously linked, I am wondering whether using “gaze” instead of “head” would clarify this sentence.

In section 5, I am curious to know whether authors could expand on frontal plane measures of stability. Control of balance during locomotor activities is determinant in the frontal plane, therefore the roll motion of the head may be a relevant area of research. I am also curious whether or not it could be interesting for the reach of this review to consider providing results on head stability and gaze fixation from studies which used external perturbations such as change in ground characteristics, or unpredictable support like a perturbation platform or ridding a horse?

All results provided in this review come from laboratory experiments, which may question how action-perception mechanisms work in a more ecological situation such as running outside of the lab. Maybe consider a short discussion of this later point?

In section 6, I recommend the authors to re-emphasize on the coordination mechanisms that have been highlighted in previous sections such as, for instance, the importance of the stance phase, or the mechanisms that help reducing shock waves (increased knee flexion, increased stride frequency and reduction of aerial time).

To me, the text presented in l. 365 to 368 refers to results from Lim et al. study and could be included in section 4.

I suggest the authors to add a final section to their manuscript, where they summarize their recommendations for futures lines of research in this area. On what should we focus in the future?

Author Response

Reviewer 2

Summary

This literature review summarizes results from studies investigating behavioral mechanisms used to optimise head stability during running in humans. The purpose of the study is to focus on how visual perception influences locomotor coordination and reduction of ground-related impacts as well as how changes in joint-segment coordination may influence head stability and visual perception. After an Introduction of the Perception-Action framework and demonstrating that reflexive mechanisms engaged in head stabilization may not be enough to explain the whole head stabilization process, authors describe results from three main studies. The main results from each study represents a whole section and is presented in perspective with results from other studies. A paragraph on how coordination contributes to head stability follows, and a final paragraph summarizes the knowledge we gain from this review.

General comment

The work done by the authors is very good. The paper is very well written, problematics and results are presented with insight and perspective. This review will surely be an important contribution to a developing field of research and may become an important reference paper in the future. I do, however, have two main comments and a few questions and recommendations.

Main comments

Authors did not explain how they selected the different papers they chose to present in this literature review. Did you select them on the basis of keywords? Which database(s) did you use? Were some papers excluded? If yes, can you provide the methodology that has been used to select or disregard papers from this manuscript?

The purpose of this paper was to review a thrust of research on the coupling relationship between vision and locomotion research conducted in our laboratory. The main papers (Busa et al., Lim et al. and Lim et al.) that we reviewed were conducted in our laboratory. This is stated in the paragraph at the end of the introduction (Lines 96-99). All other papers in the reference list were used to support/contradict the main findings of our papers.

It may be concerning that authors refer to data from a paper that is still “under review” (and therefore comes from the same research group). Although I trust the honesty of the authors for presenting these results, changes may be recommended by the experts currently reviewing the other paper. Wishing you the best of luck for publication of the other paper, there is no way for me to verify the results and design of the experiment of Lim et al. [59].

We agree that referring to a paper under review may be considered problematic. The material in this paper will be presented at a major conference soon and the abstract has already been published. Thus, we reference the abstract and not the paper.

Remarks / recommendations

Since the authors chose the Action-Perception framework to position their work, I was wondering if the current title may not be misleading as authors do not refer to the perception concept. Biomechanicians may see something only referring to segmental coordination. Maybe consider including perception somehow?

We have changed the title to “Locomotor Coordination, Visual Perception and Head Stability During Running” as suggested.

In the Introduction, authors relate to perception as “individual’s perception of the environment” (l. 37-38) and to “perception-action coupling under different visual task constraints” (l. 42-43). Although it is clear that the authors’ point is to drive the reader towards visual perception that indeed contributes to perception of the environment, the whole perception mechanism is not limited to the environment nor to vision only. Perception of our own bodies may also influence our actions (and vice-versa), while several sensory mechanisms could be at work additionally to visual feedback. Several references to other sensory systems are actually made throughout the paper. Therefore, I only recommend adding a sentence or two to clarify this and then dive into what authors will focus on in the review. According to this remark, the schematic illustration of the Perception-Action coupling could also be clarified with something like “perceptual information from the environment and the human body” coming with the right-side arrow.

Thanks for catching this. We have changed the text in lines 3-41 as follows: “This coupling involves a continuous, cyclic relationship between information from the body (proprioception) and the environment and the necessary adjustments of the body from which a stable pattern of behavior occurs.” We have also made the change in Figure 1.

I recommend clarifying why authors chose to focus on the three specific studies they are presenting within individual sections and why they chose to separate each study from one another. At first glance, these three studies may look quite similar. One sentence or two to clarify the authors’ idea about why each study is different from another may help to understand the organization of sections 2, 3 and 4.

The purpose of this paper was to review a thrust of research on the coupling relationship between vision and locomotion research conducted in our laboratory. The main papers (Busa et al., Lim et al. and Lim et al.) that we reviewed were conducted in our laboratory. This is stated in the paragraph at the end of the introduction. All other papers in the reference list were used to support/contradict the main findings of our papers. In the purpose paragraph of the paper we state: “We will consider three studies that were conducted in our laboratory that attempted to answer the questions relating visual precision during altered stride frequency conditions (i.e., changing stride length to cause different foot-ground collisions).” (Lines 96-99)

In section 2, l. 151-152, it is mentioned that studies from Lucas-Cuevas et al. [28] and Mangubat et al. [29] support the view and the results of Hamill et al. study [20]. Given the content of both these studies, it questions why authors decided the study from Hamill et al. was the “main” study in this paragraph. A little clarification may help the reader understand why the study from Hamill et al. is the main one referred in this paragraph.

The Hamill et al. [20] paper suggested that the body attenuated impact shock to preserve head stability for the purpose of maintaining a constant visual field. This was supported by the studies of Lucas-Cuevas et al. [28] and Mangubat et al. [29].

  1. 242, are authors talking about head stability demands or gaze stability demands? Given the different conditions presented in Lim et al. study, they constrained gaze orientation over head orientation. If both are obviously linked, I am wondering whether using “gaze” instead of “head” would clarify this sentence.

We have changed this terminology to use ‘head-gaze’ rather than ‘head’ or ‘gaze’ alone.

In section 5, I am curious to know whether authors could expand on frontal plane measures of stability. Control of balance during locomotor activities is determinant in the frontal plane, therefore the roll motion of the head may be a relevant area of research. I am also curious whether or not it could be interesting for the reach of this review to consider providing results on head stability and gaze fixation from studies which used external perturbations such as change in ground characteristics, or unpredictable support like a perturbation platform or riding a horse?

The authors agree with the reviewer’s comment that frontal plane measures are relevant to this topic. However, it is obvious that execution, adjustment, and coordination of the movement during locomotion occur predominantly in sagittal plane and followed by transverse plane and frontal plane. While it is possible to expand the analysis to the frontal plane, we decided to limit the analyses in two main planes of motion (i.e., sagittal and transverse plane) not only because the volume of the manuscript, but also because the consistency with previous literature, which have focused on the movements in two planes.

Similarly, the reviewer’s comment on the expandability of the results to the situations where external perturbations are applied is out of the scope of the current review in the context of locomotion, and therefore, could be an interesting future study/review.

All results provided in this review come from laboratory experiments, which may question how action-perception mechanisms work in a more ecological situation such as running outside of the lab. Maybe consider a short discussion of this later point?

We have added the following text in the summary section (Line 384-392): “In our laboratory studies, all participants ran on a treadmill which is a reasonable surrogate for over-ground running.  However, the perception of speed, movement or optic flow may not be the same.  Therefore, repeating our laboratory studies over-ground rather than on a treadmill may present a more ecological setting for interpreting the coupling of movement and perception and the control of head stability.  In addition, the methods and techniques presented in this review can be applied to those with visual impairments and/or gait asymmetries, where different demands on head control may exist or a diminished capacity to modulate coordination patterns in response to different visual task constraints.”

In section 6, I recommend the authors to re-emphasize on the coordination mechanisms that have been highlighted in previous sections such as, for instance, the importance of the stance phase, or the mechanisms that help reducing shock waves (increased knee flexion, increased stride frequency and reduction of aerial time).

We added the following to the Summary section: “Changes in coordination mechanisms can also attenuate the impact from foot-ground collisions.  During the stance phase of locomotion, a shock wave caused by a foot-ground collision can also be attenuated by increased knee flexion of the support limb or increasing stride frequency.” (Lines 357-360)

To me, the text presented in l. 365 to 368 refers to results from Lim et al. study and could be included in section 4.

We added this information to the Summary section for emphasis since it is also mentioned in section 4.

Lines 365-368 are about the coordination analysis results (HMS paper). But, in section 4, we summarized the kinematics and shock attenuation results (JOB paper). Based on our logic (i.e., Line 317-320), it seems better to not mention the coordination results in section 4.

I suggest the authors to add a final section to their manuscript, where they summarize their recommendations for futures lines of research in this area. On what should we focus in the future?

We have added We have added the following text in the summary section (Line 384-392): “In our laboratory studies, all participants ran on a treadmill which is a reasonable surrogate for over-ground running.  However, the perception of speed, movement or optic flow may not be the same.  Therefore, repeating our laboratory studies over-ground rather than on a treadmill may present a more ecological setting for interpreting the coupling of movement and perception and the control of head stability.  In addition, the methods and techniques presented in this review can be applied to those with visual impairments and/or gait asymmetries, where different demands on head control may exist or a diminished capacity to modulate coordination patterns in response to different visual task constraints.”